# OpenReview forum: "The Last Iterate Advantage: Empirical Auditing and Principled Heuristic Analysis of Differentially Private SGD"
_ICLR.cc/2025/Conference — ICLR 2025 Poster_

### Official Review · Reviewer_MfJb · 2024-10-31

**Soundness:** 3
**Presentation:** 2
**Contribution:** 2
**Rating:** 5
**Confidence:** 4

**Summary:**

The paper computes heuristic differential privacy parameters $\varepsilon$ and $\delta$ for releasing the last iterate of the DP-SGD algorithm. The proposed heuristic relies on computing the DP upper bound under linear loss function, which is always smaller than the DP upper bound under worst-case loss functions.  Numerical experiments show interesting scenarios where the proposed heuristics yield estimates that are significantly smaller than the worst-case DP upper bound. Experiments on image and language datasets confirm that the heuristic estimates lie between the DP upper bound and the lower bounds obtained via privacy auditing. Finally, the authors provide examples where the proposed heuristics underestimate the privacy loss.

**Strengths:**

- A systematic investigation of using heuristic privacy estimates to study the last-iterate privacy loss of the DP-SGD algorithm, including derivations, numerical experiments, comparison with auditing experiments, and analysis of counterexamples where the proposed heuristics underestimates the privacy loss.

- Numerical experiments show that when the sampling rate is small, the proposed heuristic yields an estimate that is significantly smaller than the standard DP upper bound.

- Auditing experiments illustrated that interestingly, black-box gradient space attacks fail to give tight auditing lower bound for the last-iterate of DP-SGD algorithm under natural datasets, in which case the heuristic estimates and the input-space attack gives more reliable estimates for the privacy loss.

**Weaknesses:**

- The message of the paper is not fully clear -- why is such a heuristic privacy estimate realistic and useful? Specifically, (1) linear loss is not a commonly used loss function; and (2) the heuristic estimates appear to be roughly the same as DP upper bound in certain auditing experiments (Figure 2). See questions 1 and 2 for more details.


- Several terms and plots in the paper are not explained in detail and the claims require more clarification. See question 3 for more details.

**Questions:**

1. Could the authors comment on why the proposed heuristics give a realistic privacy risk estimate, given that the linear loss function is not commonly used? That is, whether and how often would the proposed heuristics overestimate the last-iterate privacy loss.

2. Besides the small sampling rate and the pathological counterexamples (as discussed in 4.3), are there other regimes where the proposed heuristic is significantly smaller than the DP upper bound? This is to understand the usefulness of the proposed heuristics.

3. Minor questions regarding clarity:
    - Line 297 - `we expect this looseness to be the result of statistic effects` -- could the authors provide error bars in Figure 3 to validate this hypothesis?
    - Figure 3 and 4: why is the standard epsilon constant under increasing the number of steps?
    - What is the exact definition of a heuristic privacy estimate? It seems to be DP's upper bound over a family of loss functions, rather than all loss functions.

---

> ### Author Response · Authors · 2024-11-26
>
> We thank the reviewer for their input. We respond to the questions below.
>
> > 1. Could the authors comment on why the proposed heuristics give a realistic privacy risk estimate, given that the linear loss function is not commonly used? That is, whether and how often would the proposed heuristics overestimate the last-iterate privacy loss.
>
> Linear loss functions are not used in practice, but our thesis is that linear losses are close to the worst case in terms of privacy for realistic losses in the hidden state/last iterate setting. Thus it makes sense to study them, as we do.
>
> The justification for this thesis is twofold: First, in the special case of full batch gradient descent it is provably true (see Appendix B). Second, as observed in previous papers on privacy auditing -- and replicated in our experimental results -- the strongest privacy auditing results are achieved by making the loss behave more like a linear loss.
>
> Our method is a heuristic -- it may overestimate the true privacy loss and it may underestimate it. We spent a great deal of effort investigating when and how this might occur.
>
> In practice, one does not know the true privacy loss. There are provable upper bounds and empirical privacy auditing lower bounds; the truth could be anywhere in between. Our heuristic offers a novel perspective. It gives a third number that is neither an upper nor a lower bound -- it should be in between and it gives some possible explanations for the gap. We believe this has value even if it's not the perfect answer people are looking for.
>
> > 2. Besides the small sampling rate and the pathological counterexamples (as discussed in 4.3), are there other regimes where the proposed heuristic is significantly smaller than the DP upper bound? This is to understand the usefulness of the proposed heuristics.
>
> DP-SGD with a small sampling rate is the setting of practical interest. In practice, datasets are large, batches are small, and training epochs are few, which all correspond to a small sampling rate. Thus we focus on this setting.
>
> > 3. Minor questions regarding clarity:
> > *  Line 297 - we expect this looseness to be the result of statistic effects -- could the authors provide error bars in Figure 3 to validate this hypothesis?
>
> For privacy auditing we aim to give a lower bound on the true privacy loss (with 95% confidence). Thus we are effectively reporting the lower end of the confidence interval already.
>
> The reason we believe there is a big gap between the auditing result and our heuristic for $T=1$ step is that with a low sampling rate ($q=0.01$) we will only see the canary in 1% of runs. There's just not enough data to perform a high-confidence attack. We will clarify this sentence.
>
> > * Figure 3 and 4: why is the standard epsilon constant under increasing the number of steps?
>
> We adjust the noise scale to keep this constant. So more steps correspondingly has more noise per step.
>
> > * What is the exact definition of a heuristic privacy estimate? It seems to be DP's upper bound over a family of loss functions, rather than all loss functions.
>
> There is no exact definition. This is why we use the term "heuristic".

---

### Official Review · Reviewer_YKem · 2024-11-01

**Soundness:** 4
**Presentation:** 4
**Contribution:** 3
**Rating:** 8
**Confidence:** 4

**Summary:**

This paper derives an exact privacy analysis of DP-SGD for linear models (referred to as the heuristic) when only the last iterate is released. The authors also compare existing empirical privacy auditing methods with the exact heuristic across different regimes, including image classification tasks and language models.

**Strengths:**

This paper introduces a novel approach (named heuristic) to assess the performance of existing empirical evaluations of DP-SGD's privacy guarantees. While empirical evaluation provides a lower bound on the privacy budget of DP mechanisms, it may underestimate the true privacy loss. By comparing it with the heuristic bound for linear models, if the empirical audit bound is loose, it is unreasonable to expect it to hold tightly for more complicated non-linear (or even non-convex) models.

**Weaknesses:**

The main weakness of this paper is its limited theoretical contribution. In my view, the heuristic bound (Theorem 1) offers a simplified analysis of existing convergence privacy analysis for (strongly) convex loss functions (cf., the literature referenced in the introduction of this paper, which uses both RDP and $f$-DP). Additionally, the linear model is relatively simple, as the last-iterate output has a closed-form representation. Therefore, I propose the following questions or modifications:

1. $\textbf{Linear Probing as a Benchmarking Method:}$ Linear probing is commonly used for benchmarking when privately fine-tuning a foundation model. Therefore, the proposed heuristic for linear models could be beneficial when fine-tuning the last layer (linear probing) of a foundation model. Besides the well-known paper by De et al., the following recent studies may provide strong support for the utility of linear models in private fine-tuning, potentially strengthening the story for using heuristics for linear models.

Differentially Private Image Classification by Learning Priors from Random Processes. Tang et al., NeurIPS'23.

Neural Collapse Meets Differential Privacy: Curious Behaviors of NoisyGD with Near-perfect Representation Learning. Wang et al., ICML'24.

2. $\textbf{Extension to Convex Loss Functions:}$ Could this heuristic bound be extended to convex loss functions? I understand that the existing convergence bounds for DP-SGD involve complicated constants that are challenging to specify in practice, but a potential comparison between empirical auditing bounds and the theoretical upper bound under convex loss would be more convincing than in the linear case.

3. $\textbf{Comparison with Privacy Auditing Bound using privacy profiles: }$  It appears that the comparison with the privacy auditing bound is based on a specific value of $(\epsilon, \delta)$. What about examining the entire $(\epsilon, \delta(\epsilon))$-curve (or equivalently, the ROC curve or the type I/II error trade-off curve)? My question arises because, even if the privacy auditing bound might be (nearly) tight for specific values of $(\epsilon, \delta)$, it may not remain tight across the entire curve.

**Questions:**

See the weakness part.

---

> ### Author Response · Authors · 2024-11-25
>
> We thank the reviewer for their time and comments. We respond to the comments below.
>
> > The main weakness of this paper is its limited theoretical contribution.
>
> The purpose of our paper is to introduce a novel viewpoint (heuristic analysis) to accompany the existing lines of work on provable differential privacy guarantees and privacy auditing. We acknowledge that the technical novelty is limited (although we believe that our empirical evaluations & counterexamples are significant technical contributions). But our hope is that the conceptual contribution is of interest to the ICLR audience.
>
> > 1. **Linear Probing as a Benchmarking Method:** Linear probing is commonly used for benchmarking when privately fine-tuning a foundation model. Therefore, the proposed heuristic for linear models could be beneficial when fine-tuning the last layer (linear probing) of a foundation model. Besides the well-known paper by De et al., the following recent studies may provide strong support for the utility of linear models in private fine-tuning, potentially strengthening the story for using heuristics for linear models.
>
> We are not sure we understand this suggestion. Fine-tuning the last layer of a foundation model involves training a linear model, but the loss function is still nonlinear due to the softmax and cross entropy loss.
>
> > 2. **Extension to Convex Loss Functions:** Could this heuristic bound be extended to convex loss functions? I understand that the existing convergence bounds for DP-SGD involve complicated constants that are challenging to specify in practice, but a potential comparison between empirical auditing bounds and the theoretical upper bound under convex loss would be more convincing than in the linear case.
>
> Our heuristic analysis heavily relies on the simple structure of DP-SGD for linear loss functions.
> In our counterexamples section we generalized our analysis to quadratic loss functions/regularizers, which we think is a good proxy for general convex losses. This was already nontrivial.
> Thus it seems difficult to generalize to arbitrary convex losses.
>
> Unfortunately, it is difficult to extract directly comparable bounds from the related work on convex loss functions. These works are theoretical in nature; in particular, many have unspecified constants and the results often only apply to certain asymptotic regimes. Their bounds also depend on parameters like the strong convexity, smoothness, and the diameter of the parameter space. And it's not clear how to specify these for a fair comparison.
>
> > 3. **Comparison with Privacy Auditing Bound using privacy profiles:** It appears that the comparison with the privacy auditing bound is based on a specific value of $(\epsilon,\delta)$. What abound examining the entire $(\epsilon,\delta(\epsilon))$-curve (or equivalently, the ROC curve or the type I/II error trade-off curve)? My question arises because, even if the privacy auditing bound might be (nearly) tight for specific values of $(\epsilon,\delta)$, it may not remain tight across the entire curve.
>
> Thanks. This is a good suggestion. We focused on $\varepsilon$ for a fixed $\delta$ because this is standard in the literature.
>
> It's worth noting that state-of-the-art auditing methods already incorporate "curve fitting" for precisely this reason. I.e., the auditing lower bounds are not tight for small $\delta$ for reasons of statistical uncertainty -- it's hard to accurately estimate small probabilities. Thus SOTA methods compute lower bounds for moderate/large $\delta$ and extrapolate these numbers to small $\delta$.
>
> One of the benefits of our heuristic is that we don't need to worry about statistical uncertainty. (Instead we need to worry about whether the heuristic is "good".)

---

### Official Review · Reviewer_SieV · 2024-11-01

**Soundness:** 2
**Presentation:** 3
**Contribution:** 2
**Rating:** 5
**Confidence:** 5

**Summary:**

This paper tackles a fundamental problem in DP-SGD: the privacy analysis of the last iterate. It is well known that DP-SGD (in the centralized case) makes an artificial assumption that all intermediate iterates are published which can be observed by adversary to ease the privacy analysis through composition. New heuristic auditing methods are presented to approximate the leakage from the last-iterate. Disadvantage and failure cases of proposed methods are discussed.

**Strengths:**

The paper is well motivated and organized. Interesting examples are presented with good intuitions.

**Weaknesses:**

My primary concern is with the meaningfulness of the main result in Theorem 1.

In the proof, the authors appear to use $v_i$ to represent the $i$-th per-sample gradient, assuming $v_i$ is constant. First, I am not aware of any real-world loss function, even in the case of linear regression, that yields or can be framed to yield a constant gradient independent of model weight. Second, in practice, when $v_i$  is treated as a random variable, it is important to note that (a) the variable $v_i$  at different iterations is correlated with $v_j$  , and (b) the distribution of $v_i$ ​  changes when it is derived from a pair of adjacent datasets. Thus, when
$v_i$  is random, the proof no longer holds, as the divergence cannot simply be reduced to that between the two Gaussian mixtures derived.

This simplification also leads to non-monotonicity, as discussed by the authors. Although they propose taking the worst-case estimate across all parameter selections, this approach feels somewhat ad hoc. The absence of results demonstrating, at a minimum, the conditions under which the model in Theorem 1 can serve as a provable upper bound for the estimate makes it challenging to evaluate and fully understand this auditing method.

Additionally, can the authors elaborate on their statement about the justification for focusing on linear losses which stems from the observation that existing auditing techniques achieve the highest epsilon values? The observation itself makes sense as if every time is worst case, the privacy loss seems to be the maximal. But I do not get why this supports the reduction to the linear function,

Moreover, I believe a ground truth for the last-iterate privacy loss is missing.

**Questions:**

In general, I think there is some interesting results in the paper but more work seems to be needed.

1. Can the author comment or empirically get the truly last-iterate in some practical tasks (it is ok to just run two or three iterations) and then compare it with the estimate from the heuristic auditing proposed?

2. Can the author explain how to generalize the analysis to capture the practical random $v_i$ scenario?


minors:
1. There is overlap in the figures in Fig.1.

---

> ### Author Response · Authors · 2024-11-23
>
> We thank the reviewer for their time. We respond to their comments/questions below.
>
> > First, I am not aware of any real-world loss function, even in the case of linear regression, that yields or can be framed to yield a constant gradient independent of model weight.
>
> A constant gradient independent of model weights is precisely what a linear loss function is.
>
> However, we wish to emphasize that we are **not** assuming that realistic losses are linear. That assumption would obviously be false. As we stated in the paper, "assuming linearity is unnatural from an optimization perspective, as there is no minimizer." That is to say there is an obvious reason why linear losses never arise in practice -- the optimization procedure would not converge (unless we add a regularizer or a projection step).
>
> Our thesis is that, in terms of privacy, linear losses are close to the worst case for realistic losses.
>
> For full batch gradient descent that claim is provably true (see Appendix B). Our observation is that it also seems to be close to true even for minibatch gradient descent, at least as far as existing privacy auditing methods are concerned.
>
> > The absence of results demonstrating, at a minimum, the conditions under which the model in Theorem 1 can serve as a provable upper bound for the estimate makes it challenging to evaluate and fully understand this auditing method.
>
> It would be great if we could give provable general-purpose upper bounds on privacy loss. But that would be a different paper.
> Our counterexamples section shows why it is difficult to translate our heuristic into a provable general-purpose upper bound.
> We hope that future work advances in this direction. And our paper helps set a target for such improved upper bounds.
>
> > Additionally, can the authors elaborate on their statement about the justification for focusing on linear losses which stems from the observation that existing auditing techniques achieve the highest epsilon values? The observation itself makes sense as if every time is worst case, the privacy loss seems to be the maximal. But I do not get why this supports the reduction to the linear function,
>
> If we could identify the true worst case pair of inputs, then we would simply analyze those and provide a general-purpose upper bound on the privacy loss for all inputs.
> While linear losses are not the true worst case (as evidenced by our counterexamples section), they seem to be close to the worst case for realistic losses. (Of course, "close" and "realistic" are open to interpretation.) Thus we propose analyzing linear losses as a heuristic.
>
> > Can the author comment or empirically get the truly last-iterate in some practical tasks (it is ok to just run two or three iterations) and then compare it with the estimate from the heuristic auditing proposed?
>
> Unfortunately, we do not understand this question.
>
> > Can the author explain how to generalize the analysis to capture the practical random $v_i$ scenario?
>
> In general all we need is a bound on the total length of the gradients I.e. $\| \sum_t v_i^t \|$ where $v_i^t$ denotes the gradient of the $i$-th example at step $t$.
>
> The difficulty in generalizing our analysis is not that the canary gradients may be random.
> What makes it difficult is that the canary gradients and the gradients of the other examples (and the regularizer) will not be independent in general, since they all interact with the model weights. Intuitively, the influence of the canary can be amplified.

---

### Official Review · Reviewer_PBks · 2024-11-04

**Soundness:** 3
**Presentation:** 3
**Contribution:** 2
**Rating:** 5
**Confidence:** 5

**Summary:**

The paper introduces a heuristic privacy analysis for DP-SGD when only the final model is released, and intermediate updates are hidden. This heuristic assumes linear loss functions and, when the assumption holds, provides a more accurate estimate of privacy leakage than standard composition-based analyses, which often overestimate privacy loss by assuming adversaries have access to all training iterates. The authors experimentally demonstrate that their heuristic closely predicts the outcomes of privacy auditing tools, serving as a practical upper bound on privacy leakage in deep learning settings (where auditing is usually expensive). They also discuss counterexamples where the heuristic underestimates privacy leakage, highlighting its limitations. By bridging the gap between theoretical upper bounds and empirical lower bounds from privacy auditing, the heuristic sets a more realistic target for both theoretical improvements and practical attacks. It offers a computationally efficient way to estimate privacy leakage, aiding in tasks like hyperparameter selection before training without the overhead of extensive privacy audits.

**Strengths:**

* The paper presents a new heuristic privacy analysis for DP-SGD when only the final model is released.
* The heuristic allows practitioners to estimate privacy leakage before training, aiding in hyperparameter selection without the computational cost and complexity of running privacy audits.
* The heuristic can serve as a benchmark for future improvements in both theoretical privacy analyses and practical attack methods, encouraging the development of stronger privacy attacks against ML models
* The authors provide a good amount of examples and intuition when the heuristic does not hold

**Weaknesses:**

* The reliance on linear loss functions is a simplification. This mismatch may limit the applicability of the heuristic to real-world models as a valid upper bound. (see the questions)
* The paper could position itself within the existing body of work on privacy auditing, particularly recent methods that perform effective audits under similar threat models. A deeper comparison would clarify the novelty and contribution of this work. (see the questions)
* While formalising the heuristic is helpful, it can be seen as a natural extension of existing observations that independent gradients are optimal for tight auditing when only the last iterate is revealed. The paper may not fundamentally add new knowledge to privacy auditing beyond this formalisation. (see the questions)

**Questions:**

* How does this work relate to other recent results in privacy auditing that perform effective audits under similar threat models? Specifically, could the authors elaborate on the novelty of their approach compared to methods presented in papers [1], [2], and [3], which make assumptions about the adversary and not the loss itself?
* Could the authors comment on the strength of the linearity assumption compared to other assumptions or heuristics used in privacy auditing? For instance, in [4], an auditing procedure for one-shot auditing is designed under the assumption that the adversary can insert a known random gradient, which can easily be extended to the blacbox setting. How does the linearity assumption compare?
* Are there practical scenarios or specific types of models where the linearity assumption approximately holds?
* Theoretical work also addresses this threat model for the non-convex case (see [5]). Is there any connection or similarity between the works?

[1]: https://arxiv.org/pdf/2405.14106
[2]: https://arxiv.org/pdf/2405.14457
[3]: https://arxiv.org/pdf/2407.06496
[4]: https://arxiv.org/pdf/2302.03098
[5]: https://arxiv.org/pdf/2305.09903

---

> ### Author Response · Authors · 2024-11-22
>
> We thank the reviewer for their comments and we respond to their questions below.
>
> > How does this work relate to other recent results in privacy auditing that perform effective audits under similar threat models? Specifically, could the authors elaborate on the novelty of their approach compared to methods presented in papers 1, 2, and 3, which make assumptions about the adversary and not the loss itself?
>
> We thank the reviewer for bringing these recent papers to our attention. (These appeared online after we wrote the literature review, but before the ICLR deadline, so we will update our paper accordingly.)
>
> The cited papers all try to improve privacy auditing in the hidden state model, by changing the power of the adversary.
> [Annamalai et al. [1]](https://arxiv.org/abs/2405.14106) allow the adversary to choose the initial model weights.
> [Cebere et al. [2]](https://arxiv.org/abs/2405.14457) allow the adversary to insert arbitrary gradient canaries.
> [Annalamai [3]](https://arxiv.org/abs/2407.06496) allows the adversary to choose the loss function.
>
> The third paper [3] is closely related to our counterexamples. Specifically, it constructs a clever loss function that keeps track of the likelihood ratio during the training process, which makes it easy to perform a powerful attack using only the last iterate.
>
> The other two papers [1,2] both support the intuition behind our heuristic in that they both construct settings that behave like linear losses. The first paper [1] chooses the model parameters such that the gradients of the non-canary examples are approximately zero. The second paper [2] inserts constant gradients -- which corresponds to linear losses.
>
> > Could the authors comment on the strength of the linearity assumption compared to other assumptions or heuristics used in privacy auditing? For instance, in 4, an auditing procedure for one-shot auditing is designed under the assumption that the adversary can insert a known random gradient, which can easily be extended to the blacbox setting. How does the linearity assumption compare?
>
> [Andrew et al. [4]](https://arxiv.org/abs/2302.03098) insert a random constant gradient. Constant gradients correspond to linear losses. So this paper also supports the intuition that linear losses are the right thing to look at.
>
> > Are there practical scenarios or specific types of models where the linearity assumption approximately holds?
>
> We wish to clarify that we are *not* assuming that realistic losses are approximately linear. That assumption is clearly not true.
> Our thesis is that linear losses are close to the worst case in terms of privacy for realistic losses.
>
> By the same token, the aforementioned papers [1,2,3,4] consider unrealistic adversaries. Our observation is that most [1,2,4] of these unrealistic adversaries could achieve the same results with linear losses; that doesn't imply that linear losses are realistic. Rather, that implies that linear losses capture a lot of the power of privacy auditing.
>
> > Theoretical work also addresses this threat model for the non-convex case (see 5). Is there any connection or similarity between the works?
>
> The work of [Asoodeh & Diaz [5]](https://arxiv.org/abs/2305.09903) is very interesting and also considers the last iterate setting, but the contributions seem tangential to our work. Specifically they rely on the probabilistic contraction of the Markov kernels under hockey-stick divergence. Linear losses do not exhibit this property because we assume an unbounded parameter domain.

---

### Author Response · Authors · 2024-12-02

We again thank the reviewers for their valuable feedback. We have responded to all of the reviewers and we hope that this has clarified their questions about the submission. If there are any further questions, we are happy to answer them.

We wish to reiterate that the high-level point of our work is to offer a novel perspective on the privacy of DP-SGD. Our approach doesn't fit the usual paradigms of provable theoretical upper bounds or empirical auditing lower bounds on the privacy leakage. That makes it hard to evaluate our contribution, but we believe that novel approaches are needed since the existing approaches seem unable to fully shed light on the privacy properties of DP-SGD.

---

### Meta-Review · Area_Chair_gbqb · 2024-12-21

**Metareview:**

The paper investigates the privacy of DP-SGD when only the last iterate is provided (and intermediate model states are hidden). It shows a heuristic analysis for linear functions which is found to be predictive of the privacy obtained via auditing on various training procedures. The heuristic is evaluated on image and language datasets and the paper shows that it consistently upper bounds the privacy loss obtained via auditing. Though the proposed technique is a heuristic and is not rigorous, I think it provides an interesting avenue for future research. There are clearly gaps between the current theoretical upper bounds in DP and results obtained via auditing, and the work could be a step towards rigorously closing it.

I also suggest that the authors incorporate the recent contemporaneous work in the related work in the revision of the paper.

**Additional Comments On Reviewer Discussion:**

The reviewer's raised several questions, including regarding relation to prior work and why the heuristic is meaningful. The comments seem to be addressed by the author's response.

---

### Decision · Program_Chairs · 2025-01-22

Accept (Poster)